# A Bayesian Approach to Unsupervised, Non-Intrusive Load Disaggregation

**DOI:** 10.3390/s22124481

**Published:** 2022-06-14

**Authors:** Luca Massidda, Marino Marrocu

**Affiliations:** CRS4, Center for Advanced Studies, Research and Development in Sardinia, Loc. Piscina Manna Ed. 1, 09050 Pula, CA, Italy; marino.marrocu@crs4.it

**Keywords:** energy disaggregation, non-intrusive load monitoring, NILM, Bayesian methods

## Abstract

Estimating household energy use patterns and user consumption habits is a fundamental requirement for management and control techniques of demand response programs, leading to a growing interest in non-intrusive load disaggregation methods. In this work we propose a new methodology for disaggregating the electrical load of a household from low-frequency electrical consumption measurements obtained from a smart meter and contextual environmental information. The method proposed allows, with an unsupervised and non-intrusive approach, to separate loads into two components related to environmental conditions and occupants’ habits. We use a Bayesian approach, in which disaggregation is achieved by exploiting actual electrical load information to update the a priori estimate of user consumption habits, to obtain a probabilistic forecast with hourly resolution of the two components. We obtain a remarkably good accuracy for a benchmark dataset, higher than that obtained with other unsupervised methods and comparable to the results of supervised algorithms based on deep learning. The proposed procedure is of great application interest in that, from the knowledge of the time series of electricity consumption alone, it enables the identification of households from which it is possible to extract flexibility in energy demand and to realize the prediction of the respective load components.

## 1. Introduction

The growth in energy demand and the increased integration of generation from renewable energy, highly dependent on weather conditions, impose great stress on the balance of power grids and on the reliability and quality of power supply [1].

The reduced control over energy production determines the need to modify consumer energy demand through the implementation of demand side management (DSM) systems [2].

In particular, demand response (DR) represents an advanced technology in demand management, allowing the users to reduce or increase their energy consumption with the aim of responding to peaks in demand or supply in the electricity market. The net result is a greater flexibility and network stability and a more efficient use of infrastructure and energy resources [3]. In exchange for this availability to modify their consumption according to the needs of the energy supplier, the customer can receive a remuneration.

DR programs, in addition to being beneficial to the operation of power grids, can also provide economic benefits to end users. The residential sector, in particular, plays a very important role in DR programs and represents 40–60% of the potential market [4]. Domestic energy consumption in Europe represents 30% of total electricity demand and is considered the third largest sector of energy use after transport and industry. Energy consumption related to heating and cooling of residential spaces accounts for half of the EU’s residential energy consumption [5].

However, DR deployment is currently limited. A recent study [6] analyzed the impact on energy markets of large-scale DR deployment for the Northern European region. The results show that among the DR options analyzed, space heating and water heating provide the highest shares of shifted loads with low costs, making them economically feasible. The study also shows that the highest values of electricity cost reduction by actively using DR are achievable for the domestic sector, with savings exceeding 8 EUR/MWh in 2030 for the case of Denmark.

Assessing DR potential is an essential prerequisite for estimating possible energy or power reduction so that DR proponents can target potential participants economically [7], design optimal incentive schemes [8], and implement optimal strategies [2,9]. In this context, estimating household energy use is a key requirement for DR program management and control techniques. In this context and with the deployment of smart meters for all consumers, load monitoring techniques have been implemented to obtain detailed and disaggregated energy consumption information for electrical appliances [10,11].

Therefore, in this paper we will focus on thermal-type loads for domestic users given their magnitude and the possibility of regulating these types of loads with automatic systems in order to extract flexibility in energy demand, and operate directly on the readings of the new-generation smart meters

Two types of monitoring can be distinguished in the literature: intrusive load monitoring (ILM), in which the energy usage of each appliance is measured separately using a dedicated sensor, and non-intrusive load monitoring (NILM) or load disaggregation, in which pattern recognition and artificial intelligence techniques are used to infer the activation status and energy consumption of appliances from the aggregate load measurement alone [12,13,14]. The intrusive approach is difficult to apply on a large scale, due to the cost of the sensors and the processing of the generated data.

The NILM technique was introduced by Hart’s pioneering work in the mid-1980s [15]. He was the first to use measured active and reactive power time series to estimate the on/off state of individual devices. Since then, the research community has proposed several approaches to disaggregation.

One way to categorize disaggregation algorithms is to divide them into supervised and unsupervised [16]. In a supervised method, the disaggregation of an unknown power signal is preceded by a training phase, where the algorithm learns to recognize individual device power signals from the aggregated signal based on available labeled data. Unsupervised, conversely, are those methods that do not use labeled data for disaggregation.

Among the supervised methods, those based on neural networks, both for high-frequency [17,18,19] and low-frequency [20,21] signals, had considerable success in recent years. The disadvantage of supervised methods is the need for labeled data, i.e., the time series of instantaneous consumption of all the appliances in the house. These data are costly to obtain, are often not public, and in general are not readily extensible to dwellings other than those in the measurement campaign.

A review of unsupervised methods for load disaggregation can be found in [22]. A common and quite successful approach is the factorial hidden Markov model (FHMM) [23,24]. The main drawback of most unsupervised methods is their requirement of a relatively high sampling rate (typically greater than 1/60 Hz), although recently the method has also been applied also to low-frequency data [25,26].

To date, there is limited research dedicated to developing load disaggregation methods using low-resolution smart meter data. In this case, the techniques attempt to separate consumption by type, with a focus on thermal type loads and base loads (non-temperature-sensitive).

Albert and Rajagopal proposed in [27] a hidden Markov model (HMM) to convert hourly kWh data into thermal states to identify suitable demand response resources using thermostatically controlled appliances. A sequential algorithm for energy disaggregation of the thermal component of the load from 30 min time resolution data is proposed in [28].

Another approach to separate a building’s heating and cooling energy consumption is to model their dependence on weather conditions [29]. Regression methods disaggregate the thermal component by regressing energy consumption on the temperature based on the breakpoint model [30] or the static equivalent thermal parameter (ETP) model [31], assuming a linear relationship between energy consumption for room air conditioning and room temperature [32]. A Fourier series-based regression method was proposed in [33] to decompose the whole-house load into a non-temperature-sensitive component described by sine and cosine functions, as well as a temperature-sensitive component modeled as a linear function of temperature. Zhao et al. [34] addressed the disaggregation problem for both 15 min and 1 h electrical measurements via the supervised K nearest neighbors algorithm. The same algorithm was used to disaggregate consumption based only on monthly cumulated values by comparison with disaggregated consumption of sample users equipped with appliance-level sub-metering [35]. Recently, Culière et al. proposed a Bayesian model of temperature-conditioned electricity consumption that allows disaggregating the heating component from the electric load curve in an unsupervised manner, based on daily consumption detected by smart meters [36].

The solutions proposed in the literature for disaggregating thermal-type loads are therefore almost always based on supervised methods, which require sub-metered datasets for model building, i.e., including not only the time series of overall household consumption but also those related to individual appliances. This represents a potential limitation due to the difficulty of generalization of such methods. Unsupervised approaches based on signal transient analysis are not particularly effective for modulated loads characteristic of a heat pump, for example. Finally, approaches based on regression of load versus temperature realize disaggregation with a daily granularity, and do not achieve a temporal detail that would be useful for a characterization of a consumer from a demand response perspective.

In this paper we propose a new methodology, unsupervised and non-intrusive, for the disaggregation, with hourly resolution, of the electrical load of a domestic user starting from the measurements of low-frequency energy consumption obtained from a smart meter and environmental context information. That approach allows to separate consumption into two parts: one related to thermal consumption, linked to environmental conditions (which can be modulated by exploiting the thermal inertia of buildings with home automation systems), and the remaining part related to occupants’ habits (which can be partially exploited in a demand response program through deferred activation of appliances). It is a Bayesian approach, where disaggregation is performed by exploiting the information on consumption obtained from the smart meter to update the a priori information about users’ consumption habits, to obtain a probabilistic forecast with hourly resolution of the two components. We also propose an unsupervised methodology for the prior calculation with a technique derived from approximate Bayesian computation (ABC). Finally, we apply the proposed method to the disaggregation of the loads of a house for which sub-metering data are available, thus allowing us to verify the achievable accuracy.

In the remainder of the paper we will first present the method in Section 2; the case examined will be described in Section 3, where the metrics used to calculate the accuracy will also be presented. In Section 4 we will present the results and discuss them in Section 5 before presenting our conclusions.

## 2. Method

In general, the electric consumption of a house can be decomposed into two components that we name base load and thermal load. The thermal load includes electrical demand related to heating and cooling and is characterized by seasonal variability during the year as well as by daily and weekly variability related to the habits of the occupants of the house. The base load includes all other elements of energy consumption and it is not directly dependent on weather conditions, but only on the activities of the residents. In general, it is characterized by a daily and weekly variability and not by an annual seasonality.

Given the total energy consumption of a household for a period of at least one year with a time resolution of one hour, denoted by *l*, function of time *t*, we call $l_{b}$ and $l_{t}$ the basic and thermal components. We can write
(1)$$l\left(t\right)=l_{b}\left(X_{c}\right)+l_{t}\left(X_{c},X_{w}\right)+\epsilon$$

The base load depends solely on calendar-type variables which we denote by $X_{c}$, such as the time of day, day of the week, and any holiday periods, while the thermal load also depends on meteorological-type variables, such as outdoor temperature, solar irradiance, and ground-level windiness, for example.

We simplify this schematization further and assume that the base load component depends exclusively on the time of day *h*, while the thermal component also depends on the average outdoor temperature *T*. Thus, the total load will depend on the hour of day *h* and day *d* of the period considered primarily through the temperature T(d):(2)$$l\left(h,d\right)=l_{b}\left(h\right)+l_{t}\left(h,T\right)+\epsilon$$

All the factors not considered in the decomposition determine the error ϵ.

### 2.1. Base Load Calculation

The problem of load disaggregation cannot be solved by examining one piece of data at a time in the time series, but must be approached by somehow exploiting knowledge of the annual cycles, and making assumptions about the composition of the load itself.

We then consider the daily average load $\bar{l}$; this will vary throughout the year, depending on the environmental conditions outside the home.
(3)$$\bar{l}\left(d\right)\approx \frac{1}{24}\sum_{h=0}^{23}l_{b}\left(h\right)+\frac{1}{24}\sum_{h=0}^{23}l_{t}\left(h,T\right)$$
(4)$$\bar{l}\left(d\right)\propto \bar{l_{t}}\left(T\left(d\right)\right)$$

With the assumptions made, the average daily consumption will depend only on the thermal load. The total daily load will have minimum values when the outdoor temperature is mild and there is no need to activate heating or cooling systems.

From the annual profile of average daily consumption, it is possible to identify the periods of the year when the outdoor temperature is mild and the average daily load is minimal; we denote such days by $d_{mild}$. We assume the thermal load to be null on these days.

Therefore, the probability distribution of the base load, depending only on the time of day, will be equal to that of the total load on days when the temperature is mild.
(5)$$p\left(l_{b}\right)\left(h\right)=p\left(l\right)\left(h,d\right)\;\mathrm{if}\;d\in d_{mild}$$

### 2.2. Thermal Load Calculation

Once the probability distribution of the base load has been estimated, the thermal load $l_{t}$ can be obtained from the knowledge of the total load and from the Bayes’ rule:(6)$$p\left(l_{t}|l\right)=\frac{p\left(l|l_{t}\right)\;p\left(l_{t}\right)}{p(l)}$$
where $p(l_{t}|l)$, the posterior distribution, is the probability of having a thermal load $l_{t}$ given the total load *l*; $p(l|l_{t})$ is the probability of having a total load *l* when the thermal load $l_{t}$ is known, called likelihood; $p(l_{t})$ represents the a priori (prior) distribution of the heat load $l_{t}$, without knowing the actual value of the total load *l* at a given time; finally, p(l) is the normalization factor, the probability that a given value of the total load *l* will occur for any value of $l_{t}$.

This expression allows to update the a priori estimate of the load distribution $l_{t}$ at a given instant based on the knowledge of the total load *l* at that time.

To obtain the likelihood, we estimate the probability distribution of the total load knowing the value of the thermal load and observe that this will be equal to the distribution of the base load $p(l_{b})$ translated by the given value of the thermal load:(7)$$p\left(l|l_{t}\right)=p\left(l_{b}\right)\;\mathrm{if}\;l_{b}=l-l_{t}$$

For the normalization term, it is necessary to calculate the probability of a given total load *l* for all possible values of the heat load $l_{t}$.
(8)$$p\left(l\right)=\int p\left(l|l_{t}\right)\;p\left(l_{t}\right)\;dl_{t}$$

According to the Bayesian approach, the prior represents our beliefs about the possible values assumed by the heat load, without any reference to the value of the total load at a given instant.

### 2.3. Calculation of the Thermal Load Prior

To obtain an estimate of the prior distribution of the heat load, we assume a truncated normal form for its probability distribution:(9)$$p\left(l_{t};\mu ,\sigma ,a,b\right)=\frac{1}{\sigma }\frac{\phi \left(\frac{l_{t}-\mu }{\sigma }\right)}{\Phi \left(\frac{b-\mu }{\sigma }\right)-\Phi \left(\frac{a-\mu }{\sigma }\right)}\;,$$
where
(10)$$\phi \left(x\right)=\frac{1}{\sqrt{2\pi }}\exp\left(-\frac{1}{2}x^{2}\right)\;,$$
is the probability density function of the standard normal distribution, while
(11)$$\Phi \left(x\right)=\frac{1}{2}\left(1+\mathrm{erf}\left(x/\sqrt{2}\right)\right)$$
is the cumulative distribution function of the normal. The parameters μ and σ are the mean and standard deviation of the distribution, and *a* and *b* are the bounds on the range of possible values of $l_{t}$.

The values of the upper and lower bounds can be estimated from the range of the total load, and Bayes’ rule can be applied to estimate the remaining parameters μ and σ:(12)p(μ,σ|l)∝p(l|μ,σ)p(μ,σ),
where p(μ,σ|l) represents the probability density function of the two parameters given the values of the total load, p(l|μ,σ) is the probability density of the resulting total load, given the parameters of the thermal load distribution, and represents, in a probabilistic sense, the distance between the simulation of the parametric model and the corresponding observations; finally, p(μ,σ) represents the a priori probability density function for the two parameters.

An estimate of the prior can be derived from knowledge of the total load and base load distributions, by using approximate Bayesian computation (ABC) [37,38], with a rejection algorithm, as summarized in Algorithm 1.
**Algorithm 1:** ABC rejection algorithm for thermal load prior computation
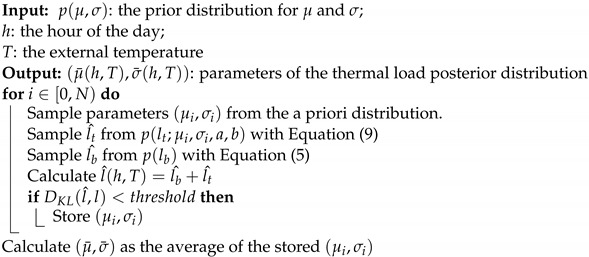


For a set of parameters (μ,σ) sampled from the prior distribution, the algorithm simulates the distribution of the total load and rejects the samples for which the distance between the simulated and observed distribution, measured by the Kullback–Leibler divergence [39], is larger than a threshold. The average of the remaining samples gives the best estimate of parameters of the prior distribution for the thermal load.

### 2.4. Disaggregation

Applying Equation (Equation 6), it is possible to estimate the posterior probability distribution of the heat load $p(l_{t}|l)$ for all the instants of a time series of the load of a household. The disaggregated thermal load can be estimated as the expected value of the resulting probability distribution:(13)$${\hat{l}}_{t}=E\left(p\left(l_{t}\right)\right)$$
and finally, the base load component can then be calculated as
(14)$${\hat{l}}_{b}=l-{\hat{l}}_{t}$$

## 3. Experimental Setup

### 3.1. AMPds Dataset

The proposed methodology has been applied to the AMPds dataset, which is freely available and already discussed in the literature [40]. AMPds contains electricity, water, and natural gas measurements at one-minute intervals for a residence located in the Greater Vancouver metropolitan area of British Columbia (Canada) for the years 2012 to 2014, to which meteorological data measured by a station located near the house is added.

The household consists of a main house, inhabited by a family nucleus consisting of two adults, both employed, and one child, and an annex rented to an adult with full-time employment. The analysis is limited to the main house (whose consumption is referred to in the dataset as MHE). The heating, ventilation, and air conditioning system (HVAC) consists of a heat pump and a forced-air gas furnace. The heat pump cools the house in the summer and heats it in the winter. The furnace is activated when the outside temperature is too low for the heat pump to operate effectively. During data collection, the HVAC thermostat is set to a constant heating set-point of 21 °C and the cooling set-point varies between 24–26 °C.

The dataset contains measurements of the overall electrical consumption and readings from sub-meters dedicated to the main appliances in the house.

The proposed methodology is unsupervised, so the only data needed are those related to the time series of total power demand and outdoor temperatures, and the electrical consumption of the appliances is used here only to test the disaggregation accuracy of the method.

The ridgeline plot in Figure 1 shows the distribution of average daily consumption according to the external temperature. It can be seen that load is minimal for temperatures around 16 °C and increases both for higher temperatures, when the room cooling system is used, and for lower temperatures, when the heat pumps are used. It is also evident that there is a decrease in consumption below 6 °C, linked to the use of the fossil fuel heating system with which the house is equipped.

Figure 2 shows the distributions of the average daily consumption of heat pump, the ventilation for the furnace, and unmonitored loads, indicated with HPE, FRE, and UNE in the AMPds dataset. These are the load components that have a larger thermal component, with a clear preponderance of the load due to the heat pump.

It can be seen that the heat pump consumption is equal to the standby consumption between 14 °C and 18 °C; it increases as the temperature decreases until 6 °C, at the same rate the variability of daily consumption also increases. Below 6 °C, average consumption decreases rapidly until it is again equal to standby consumption for temperatures below 0 °C. There is also an increase in consumption for temperatures around 20 °C certainly linked to the use of the heat pump in cooling mode.

The electrical absorption of the furnace is certainly due to the ventilation system, these are modest minimal absorptions for mild outdoor temperature conditions, and increasing as the temperature decreases, and to some extent also for high temperatures.

Finally, unmonitored consumption is minimal and with modest variability for external temperatures above 10 °C, while it increases for lower temperatures, probably due to greater use of additional heating equipment.

The thermal load is therefore given by the sum of these two components, net of their minimum value. This subtracts the non-thermal part of the unmonitored load and the constant part of the heat pump load, likely due to standby operation and ventilation. The time series will be used to verify the accuracy of the proposed disaggregation procedure.

The base load is the sum of the rest of the loads and is obtained by subtracting the thermal load from the total. The various components thereby obtained are those shown on a daily scale in Figure 1 as a function of the outside temperature, where the relationship of the thermal load distribution with the temperature and the almost constant distribution of the base load can be appreciated.

The base, thermal, and total loads also have, of course, variability over the day that we do not show here for brevity but which will be evaluated in more detail in the results description.

### 3.2. Evaluation Metrics

Disaggregation accuracy may be measured by mean absolute error (MAE), expressed as
(15)$$\mathrm{MAE}=\frac{1}{N}\sum_{t=1}^{N}\left|l\left(t\right)-\hat{l}\left(t\right)\right|,$$
where *N* denotes the total number of measurements of the load *l* and its estimate $\hat{l}$ at instants *t*. The value of MAE can be evaluated at different levels of temporal load sampling, MAEh denotes the error on the hourly load, MAEd the error on the daily consumption, and MAEm the error on the monthly average consumption. The relative mean absolute error (rMAE) is obtained from the ratio between the MAE and the average value of the total load on which the disaggregation is performed.

Following what was proposed in [41], we introduce metrics for classification: the energy-based recall *R*, which measures the part of the energy consumption that has been classified correctly, and the precision *P*, which measures the amount of power assigned to a device that actually belongs to it:(16)$$P_{i}=\frac{\sum_{t=1}^{N}min\left({\hat{l}}_{i}\left(t\right),l_{i}\left(t\right)\right)}{\sum_{t=1}^{N}{\hat{l}}_{i}\left(t\right)}$$
(17)$$R_{i}=\frac{\sum_{t=1}^{N}min\left({\hat{l}}_{i}\left(t\right),l_{i}\left(t\right)\right)}{\sum_{t=1}^{N}l_{i}\left(t\right)}$$
where $l_{i}\left(t\right)$ represents the power actually absorbed at instant *t* by the appliance *i*, and ${\hat{l}}_{i}\left(t\right)$ its estimate.

The score *F*1 is obtained from the geometric mean of precision and recall:(18)$$F1_{i}=2\frac{P_{i}R_{i}}{P_{i}+R_{i}}$$

The probabilistic accuracy of the disaggregation can be measured with the continuous ranked probability score (CRPS), defined as follows [42]:(19)$$\mathrm{CRPS}=\frac{1}{N}\sum_{t=1}^{N}\int_{-\infty }^{\infty }{\left(F\left(t,x\right)-\hat{F}\left(t,x\right)\right)}^{2}\;dx,$$
where F(t,x) is the cumulative distribution function (CDF) of the probabilistic prediction at time *t*, while $\hat{F}\left(t,x\right)$ is the CDF of the observations. Note that the CPRS coincides with the MAE for a deterministic prediction [43]. Low values of CRPS indicate good performance.

## 4. Results

An estimate of the base load and its distribution is obtained by first identifying the days $d_{mild}$ when the thermal load is minimum. In this dataset, the thermal load can be considered null on days in which the average outdoor temperature is in the range [15 °C, 17 °C], as also evident from the total load plot in Figure 1. The base load distribution is assumed to be equal to the consumption on these days and a function of the hour of the day alone. Its measured distribution is shown in Figure 3, where it is possible to appreciate the variations in the distribution of the baseload during the day. In the night hours, the energy consumption is low but also not very variable. In the early morning, there is a strong increase in consumption and in its uncertainty; during the day, consumption decreases but the uncertainty is higher than that found in the night hours. In the late afternoon, there is an increase in energy consumption associated again with a strong uncertainty on the value. Finally, consumption increases again in the evening hours before dropping at 23 h.

Figure 3 also shows the prior distribution as calculated from Equation (Equation 5). The comparison between the estimated and measured distributions shows a substantial agreement for the estimation of the average values of the load and an excellent overlap of the probability densities, and a greater uncertainty in the estimate for the early morning hours. Measures of estimation accuracy obtained with this prior distribution confirm the impression provided by the image, as reported in Table 1. As expected, the disaggregation accuracy improves as the period of aggregation increases, but even at hourly resolution, relatively good estimation is obtained for both deterministic and, especially, probabilistic prediction, as evident from the CRPS value.

With an estimate of the base component, it is then possible to obtain the prior for the thermal component of the home’s consumption using approximate Bayesian computation, as described in Section 2.3. The resulting distribution is a function of both temperature and time of the day; Figure 4 shows the comparison between the estimated and actual distributions of the thermal loads for each hour of the day for days when the average outdoor temperature was between 5 °C and 7 °C.

Table 1 shows the mean accuracy of the estimate of the thermal load component with the prior alone.

From the estimated priors for the base and thermal components and from the knowledge of the total load, the posterior probability distribution of the thermal load can be obtained.

In Figure 4, one can see the improved overlap between the measured and predicted posterior distributions, with a clear improvement with respect to the prior.

From the estimated priors for the base and thermal components and from the knowledge of the aggregate load, the posterior probability distribution of the base load can be obtained. The posterior distributions in Figure 3 show minimal visible improvement with respect to already good prior estimation, but a significant improvement of the disaggregation performance can be appreciated by the results in Table 1.

Very good performance for estimation with the posterior distribution with a significant improvement over the prior estimation is evident. Accuracy is highest as mentioned for long aggregation periods, such as month or day, but is also particularly effective for one-hour aggregation, thus achieving the goal of this paper. The performance for posterior estimation of the thermal and baseline part is clearly identical by construction, as per Equation (Equation 14).

In Figure 5, we show some examples of application of the algorithm for different periods of the year, characterized by different values of the average outdoor temperature. The plots show the total load and the effective thermal load; the thermal load estimates obtained with the proposed procedure are also shown. Overall, a good accuracy can be noticed for all the periods considered. The procedure appears to be effective both in identifying the switching on of heating and cooling systems and in estimating the value of the power absorbed by these systems, even at hourly resolution, with a remarkable accuracy for a totally unsupervised procedure.

## 5. Discussion

In order to benchmark the proposed approach against other methodologies, the accuracy obtained must be compared with other results in the literature. We did not find any work addressing the problem of the disaggregation of the entire thermal load on this dataset. The attention is often limited to white appliances. However, some work has addressed the disaggregation of the load due to the heat pump, which constitutes almost the entire thermal load and that in fact represents an excellent term of comparison.

The dataset is related to a single house, so all supervised methods, such as those based on deep learning, have high accuracy but are unrealistic, as they use part of the dataset for training of the model, and then fail the non-intrusiveness of the method. In order to properly evaluate the accuracy of a supervised method in non-intrusive disaggregation, it is necessary that the model training be conducted on a labeled dataset that does not include the house on which the test is performed. The most correct performance comparison in this case should therefore be performed with unsupervised methods, such as those based on combinatorial analysis and the hidden Markov model (HMM) technique.

In [41], a deep learning method based on a denoising auto encoder type network was compared with the AFAMAP technique based on the HMM technique. The value of F1 found with the supervised method was 64.5% while with the unsupervised method the value of the same metric resulted 54.6%, whereas with our technique the value of this measure of accuracy stands at a value of 92.0%, therefore higher than both the unsupervised and the supervised approaches. The same group obtained higher accuracies with supervised methods [44], but still worse then the present results.

In [34], a disaggregation with a kNN method was performed for the hourly data, obtaining an MAE value equal to 224.2 W for the hourly consumption of the HVAC system in the examined dataset. In addition, in [45], a supervised method based on convolutional and recurrent networks was compared with unsupervised methods, obtaining, for the heat pump of this dataset, an MAE value for the hourly consumption equal to 249.16 W with a combinatorial optimization type method and 121.69 W with a factorial hidden Markov model method. The supervised method allowed, as expected, to reach higher accuracy with an MAE value that drops to 94.79 W. Further improvements were obtained in [46], where a supervised approach based on generative adversarial network was used, by which the disaggregation of the heat pump was achieved with an MAE of 84.9 W. With our methodology, we obtained an hourly resolution MAE of 107.53 W, which is lower than the values obtained with unsupervised methods and close to the errors that can be obtained with supervised methods.

## 6. Conclusions

We presented an unsupervised procedure for the disaggregation of the electrical consumption of a house, in the base and thermal components, respectively, independent and dependent on external environmental conditions.

The procedure, of probabilistic type, exploits Bayes’ rule to obtain the estimate of the load components starting from the knowledge of the aggregate load at a given time and from the prior probability distributions of the two components, calculated through the analysis of the annual energy cycle and through an approximate Bayesian computation algorithm. The procedure was designed so that it can be applied automatically for a generic house for which the energy consumption time series with an hourly resolution is available for at least one year, in order to appreciate seasonal variations.

The procedure was then applied to the literature case of a house for which the data are available for the consumption of individual appliances, together with the aggregate household total load. The accuracy achieved in the disaggregation of loads was very good, not only for daily and monthly energy but also for the hourly resolution data, with higher performance than other unsupervised methods and comparable to those obtained with supervised algorithms based on deep learning, which require the knowledge of the consumption of individual appliances for the training of the model.

Through the introduction of additional information in the estimation of the a priori distributions of the loads it should possible to further improve the accuracy of the method. It will also be interesting to assess the applicability of this Bayesian approach for other types of loads and other types of users. In the continuation of this research, we will apply the methodology to other types of dwellings and to commercial-type buildings; we also intend to test its applicability to other types of loads from which flexibility can be automatically extracted, in particular, those related to electric vehicle charging and electric water heater operation.

## Figures and Tables

**Figure 1 sensors-22-04481-f001:**
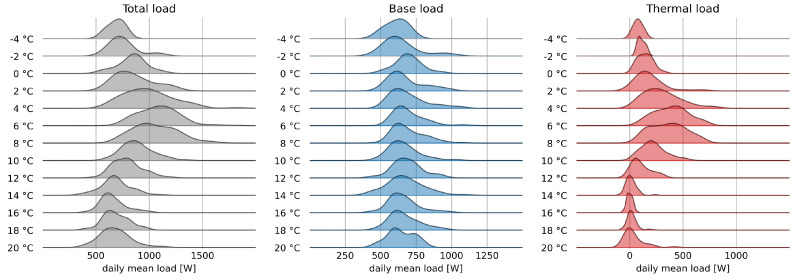
Ridgeline plot of average daily total, base, and thermal load distribution as a function of mean outdoor temperature.

**Figure 2 sensors-22-04481-f002:**
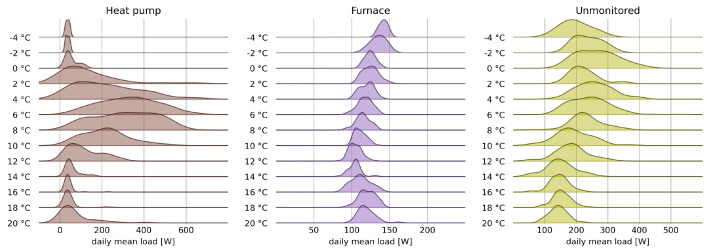
Ridgeline plot of average daily load distribution for heat pump, furnace, and unmonitored loads as a function of mean outdoor temperature.

**Figure 3 sensors-22-04481-f003:**
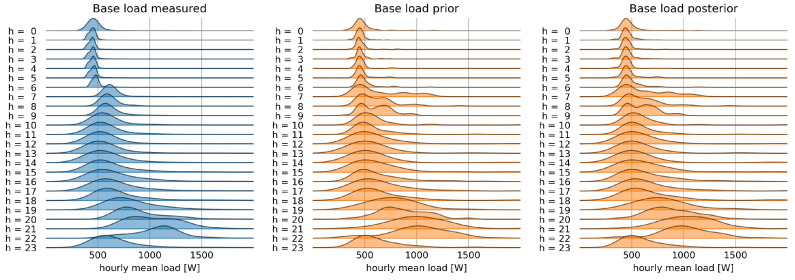
Ridgeline plot of distribution of the measured base hourly load, and of its prior and posterior distributions, grouped by the hour of the day.

**Figure 4 sensors-22-04481-f004:**
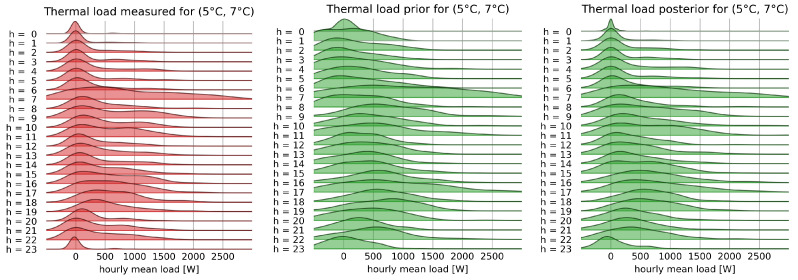
Ridgeline plot of distribution of the measured thermal hourly load, and of its prior and posterior distributions, grouped by the hour of the day, when the average outside temperature is in the range [5–7 °C].

**Figure 5 sensors-22-04481-f005:**
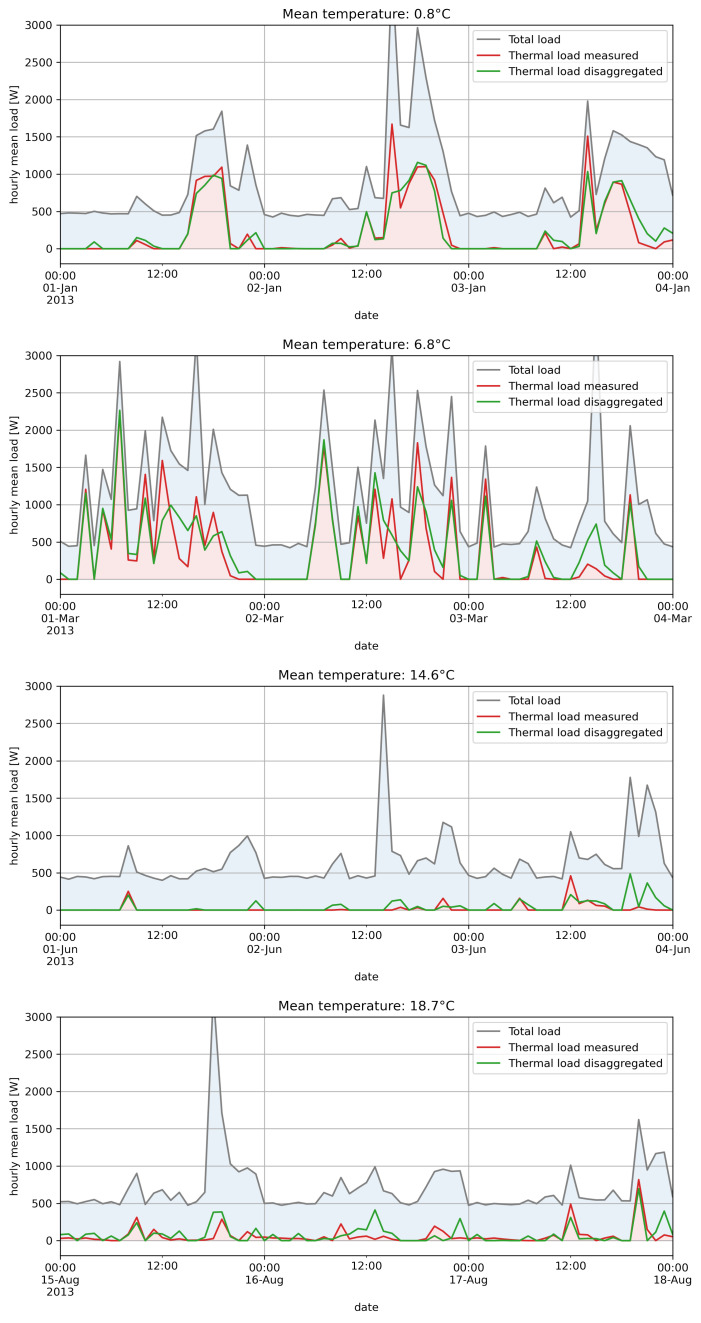
Load disaggregation examples for different periods in a year, with different values of the outdoor mean temperature.

**Table 1 sensors-22-04481-t001:** Disaggregation accuracy results for the AMPds dataset.

	Base Load		Thermal Load	
	Prior	Posterior	Prior	Posterior
MAEh (W)	162.39	107.95	215.07	107.95
rMAEh (%)	18.88	12.55	25.00	12.55
Precision (%)	88.93	93.19	88.93	93.19
Recall (%)	86.83	90.68	86.83	90.68
*F*1 score (%)	87.86	91.92	87.86	91.92
CRPSh (W)	121.96	87.81	163.80	87.81
MAEd (W)	85.30	52.24	86.91	52.24
rMAEd (%)	9.92	6.07	10.10	6.07
MAEm (W)	25.44	23.13	40.14	23.13
rMAEm (%)	2.96	2.69	4.67	2.69

## Data Availability

Not applicable.

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
