# Peer review of "A Bayesian Approach to Unsupervised, Non-Intrusive Load Disaggregation"

_sensors, 2022, doi:10.3390/s22124481_

Round 1

Reviewer 1 Report

No comments.

Author Response

Thank you very much for your appreciation of this work.

Reviewer 2 Report

The paper deals with a topic of great academic but also industrial interest. It is well written and structured. I have a few comments to make:

a. Although the references list is big, there are lots of recent NILM approaches and implementations that are not mentioned in this paper. One example is the following paper that deals with scalable NILM approach: 

Athanasiadis, Christos, et al. "A scalable real-time non-intrusive load monitoring system for the estimation of household appliance power consumption." Energies 14.3 (2021): 767.

b. Would such an implementation face difficulties when other long-lasting energy-intensive events occur in a house? E.g. an EV being charged?

c. Could the implementation be used to seperate other categories with each other (instead of hvac and others)?

Author Response

The paper deals with a topic of great academic but also industrial interest. It is well written and structured. I have a few comments to make:

We would like to thank you very much for your interest in and appreciation of this work and for the comments we respond to below, which allowed us to better clarify the potential of the approach in the new draft. The changed text is printed in red.

a) Although the references list is big, there are lots of recent NILM approaches and implementations that are not mentioned in this paper. One example is the following paper that deals with scalable NILM approach:

Athanasiadis, Christos, et al. "A scalable real-time non-intrusive load monitoring system for the estimation of household appliance power consumption." Energies 14.3 (2021): 767.

We were not familiar with this work, we thank you for the recommendation, it is a different approach from ours but certainly useful to give a more complete picture of the state of the art.

b) Would such an implementation face difficulties when other long-lasting energy-intensive events occur in a house? E.g. an EV being charged?

Our proposed method does not exploit the presence of transients in consumption, it is based solely on the statistical characterization of consumption, so we consider that the duration of the load itself has no influence on the methodology.

Intense loads might actually mask other load components, it is indeed a problem for other disaggregation methodologies. We believe that for disaggregation of the thermal component, with the proposed Bayesian methodology, this effect might not be relevant. We will verify in the continuation of our research and thank you for the investigative stimulus.

c) Could the implementation be used to seperate other categories with each other (instead of hvac and others)?

In principle, yes, and it is an issue we intend to explore further in the remainder of this research. It involves exploiting other context information to characterize the load, similar to what we have done for environmental conditions. In developing our work we would also like to approach loads related to electric vehicles and electric water heaters, which are of specific interest to the Project that funded this work, in addition to the thermal loads that are the subject of this paper.

Reviewer 3 Report

In this paper the authors study the problem of for disaggregating the electrical load of a household from low-frequency electrical consumption measurements obtained from a smart meter and contextual environmental information. This article proposes the electric consumption of a house can be decomposed into two components that name base load and thermal load.
Consistent power system operation is very important to both power utilities and consumers.
The accuracy of decomposing electric consumption  into components depends on many factors. In this study, it is not said for which consumers, industrial, domestic or mixed loads, the study is being carried out?
The topic is quite interesting, but I have comments. The introduction should place the proposed approach on the background of existing and known solution presented in literature. Also the importance of the research field should be stressed. What is new in this study?
In my opinion, it was advisable to use wavelet analysis to represent images of various electrical loads. And with the help of neural networks to recognize these images.
Another option for solving this problem is to divide the load into active (heating or cooling) and load with switching power supplies (TVs, computers, microwave ovens, etc.). Using this approach, one can also improve the methodology proposed by the authors.
Also, in my opinion, to confirm the correctness of the proposed methodology in the article, it is necessary to consider a specific example. It is necessary to consider a 10/0.4 kV substation with consumers of electrical energy of different capacities connected to it. Indicate the installed capacity of each consumer and the capacity of the substation itself. Then the reader will see how the proposed method is better than the existing approaches.
The presented report is at a very high scientific level. I believe that the present study has a significant scientific and applied contribution, which is strongly emphasized in the basically reporting volume. A slight clarification can be made in the abstract part, where the quality of the research can be enhanced. In the conclusions, it is necessary to describe what economic effect can be obtained by applying the methodology proposed by the authors.

Author Response

In this paper the authors study the problem of for disaggregating the electrical load of a household from low-frequency electrical consumption measurements obtained from a smart meter and contextual environmental information. This article proposes the electric consumption of a house can be decomposed into two components that name base load and thermal load.

Consistent power system operation is very important to both power utilities and consumers.

The accuracy of decomposing electric consumption  into components depends on many factors. In this study, it is not said for which consumers, industrial, domestic or mixed loads, the study is being carried out?

We thank the reviewer for his interest in our work and for his comments, which we have attempted to respond to below and which led to some specific changes in the new revision, which we have highlighted in blue text color.

The focus of the paper is on household-type loads, which account for a large proportion of electricity consumption in the European Union, and which are an area of focus for Demand Response programs. We have modified the introduction to better highlight the focus of this study.

The topic is quite interesting, but I have comments. The introduction should place the proposed approach on the background of existing and known solution presented in literature. Also the importance of the research field should be stressed. What is new in this study?

We thank you for your comment, which allows us to better detail the contribution to the state of the art that we propose. Abundant literature on the subject is cited in the introduction and in the section dedicated to the discussion of results. Since ours is an unsupervised procedure, more space has been given to such methods than to supervised methods based on neural networks, which have been particularly discussed in recent years and also addressed by us in a previous publication. We have taken the cue provided by your commentary to better detail the contribution to the state of the art made by our methodology, and the points that distinguish it from solutions already proposed in the literature.

In my opinion, it was advisable to use wavelet analysis to represent images of various electrical loads. And with the help of neural networks to recognize these images.

Another option for solving this problem is to divide the load into active (heating or cooling) and load with switching power supplies (TVs, computers, microwave ovens, etc.). Using this approach, one can also improve the methodology proposed by the authors.

The procedures you propose are certainly interesting, and discussed extensively in the recent literature. To accomplish what you propose, however, would require a high-frequency signal in order to be able to adequately characterize the signal with wavelet analysis, or through transient identification as in the second option. Such a signal is, however, generally obtainable only through the installation of dedicated electrical load measurement systems. Instead, our procedure focuses on low-frequency signals in which the consumption sampling has a one-hour rate. This is a signal that is directly extractable from new generation smart meters and almost always available to utilities. It would be of limited use to analyze such a low rate signal through wavelet analysis or transient characterization.

As mentioned in the discussion of the results, despite the low sampling rate of the source signal, and the fact that the procedure is totally unsupervised, we obtain results in terms of disaggregation accuracy comparable to supervised deep learning methods, and far superior to other unsupervised methods presented in the literature and applied to the same case study.

Also, in my opinion, to confirm the correctness of the proposed methodology in the article, it is necessary to consider a specific example. It is necessary to consider a 10/0.4 kV substation with consumers of electrical energy of different capacities connected to it. Indicate the installed capacity of each consumer and the capacity of the substation itself. Then the reader will see how the proposed method is better than the existing approaches.

Thank you for the suggestion, which is certainly interesting. We do not have the data to be able to perform such a verification. Datasets in the literature with characteristics similar to those you propose often report only total consumption, but submetered consumption data for thermal load would also be needed for verification of the accuracy of the disaggregation. What you propose will certainly constitute a development of our research, which is part of a European project on demand response for which submetering of several hundred households is planned. We therefore plan to further test our method on that dataset when it becomes available upon completion of the project in a few years.

The presented report is at a very high scientific level. I believe that the present study has a significant scientific and applied contribution, which is strongly emphasized in the basically reporting volume. A slight clarification can be made in the abstract part, where the quality of the research can be enhanced.

Thank you very much for the comment, and for the appreciations, we have extended the abstract trying to highlight the applied contribution. The following sentence has been added “The proposed procedure is of great application interest in that, from the knowledge of the time series of electricity consumption alone, it enables the identification of households from which it is possible to extract flexibility in energy demand and to realize the prediction of the respective load components.”

In the conclusions, it is necessary to describe what economic effect can be obtained by applying the methodology proposed by the authors.

Thank you for your comment, indeed the economic return obtainable from this type of analysis is the main motivation for application interest in these matters. We have taken your suggestion and extended the introduction by including a reference to the economic potential of this activity.